# Fluorescence-Guided Surgery and Novel Innovative Technologies for Improved Visualization in Pediatric Urology

**DOI:** 10.3390/ijerph191811194

**Published:** 2022-09-06

**Authors:** Irene Paraboschi, Guglielmo Mantica, Dario Guido Minoli, Erika Adalgisa De Marco, Michele Gnech, Carolina Bebi, Gianantonio Manzoni, Alfredo Berrettini

**Affiliations:** 1Department of Pediatric Urology, Fondazione IRCCS Cà Granda Ospedale Maggiore Policlinico, 20122 Milan, Italy; 2Department of Urology, Policlinico San Martino Hospital, University of Genoa, 16132 Genoa, Italy; 3Department of Urology, Fondazione IRCCS Ca’ Granda Ospedale Maggiore Policlinico, Università degli Studi di Milano, 20122 Milan, Italy

**Keywords:** fluorescence-guided surgery, 3D imaging, 3D printing, augmented reality, pediatric urology, children

## Abstract

Fluorescence-guided surgery (FGS), three-dimensional (3D) imaging technologies, and other innovative devices are rapidly revolutionizing the field of urology, providing surgeons with powerful tools for a more complete understanding of patient-specific anatomy. Today, several new intraoperative imaging technologies and cutting-edge devices are available in adult urology to assist surgeons in delivering personalized interventions. Their applications are also gradually growing in general pediatric surgery, where the detailed visualization of normal and pathological structures has the potential to significantly minimize perioperative complications and improve surgical outcomes. In the field of pediatric urology, FGS, 3D reconstructions and printing technologies, augmented reality (AR) devices, contrast-enhanced ultrasound (CEUS), and intraoperative magnetic resonance imaging (iMRI) have been increasingly adopted for a more realistic understanding of the normal and abnormal anatomy, providing a valuable insight to deliver customized treatments in real time. This narrative review aims to illustrate the main applications of these new technologies and imaging devices in the clinical setting of pediatric urology by selecting, with a strict methodology, the most promising articles published in the international scientific literature on this topic. The purpose is to favor early adoption and stimulate more research on this topic for the benefit of children.

## 1. Introduction

In the last few years, many efforts have been made to develop novel intraoperative devices and technologies that could assist surgeons in providing better visualization of vital organs, pathological structures, and small anatomical details during surgery [1].

In this regard, fluorescence-guided surgery (FGS), together with three-dimensional (3D) reconstructions and 3D printing technologies, has been increasingly used in urological settings to improve the understanding of the individual patient’s anatomy and help urologists plan personalized surgical procedures [2,3,4,5].

In addition, augmented reality (AR) technologies have been developing the broad field of image-guided surgery, providing a new surgical tool with the potential to improve patient surgical outcomes while minimizing perioperative complications [6,7].

Moreover, contrast-enhanced ultrasound (CEUS) [8] and intraoperative magnetic resonance imaging (iMRI) [9] have been increasingly adopted in clinical settings to better visualize fine anatomical structures and plan surgical procedures in real time.

All these novel imaging techniques and devices are considered the future of adult surgery, but they are also quickly approaching the surgical scenario in children [2,10,11,12,13].

This narrative review aims to illustrate the most promising techniques and the most innovative devices currently adopted to enhance the visualization of normal and pathological structures in the field of pediatric urology by applying a strict methodology to select the most promising articles published in the international scientific literature on this topic. The purpose is to disseminate their application and facilitate a quicker adoption for the benefit of children.

## 2. Fluorescence-Guided Surgery (FSG)

Fluorescence-guided surgery (FGS) is an intraoperative imaging modality that allows surgeons to visualize normal and pathological anatomical structures in real time by administering near-infrared (NIR) fluorescent dyes or fluorescently labeled molecules (Figure 1a) [14,15].

The main advantages of adopting this optical imaging technique during surgery are associated with the lack of ionizing radiation, the excellent contrast and sensitivity of the target tissue, and the high spatial resolution of even fine anatomical structures [14,15].

Regarding the fluorophores currently adopted in surgical settings, indocyanine green (ICG) is the most used for FGS applications. It is a US Food and Drug Administration (FDA)–approved water-soluble tricarbocyanine NIR-I dye (wavelength: 700–900 nm) with a well-established safety profile that is increasingly being employed during surgery in both adults and children [10,11,12].

By binding albumin, IGC is normally confined to the vascular stream and is entirely excreted into the biliary tract within a few hours after injection. The high water solubility and the fast biliary secretion make the visualization of tissue perfusion one of the most common clinical applications of ICG in pediatric urology [12,16].

Regarding optical imaging devices, several cameras have been adopted in children to detect the fluorescence signal and guide surgical procedures in real time. The Photodynamic EyeTM marketed by Hamamatsu Photonics Co (Hamamatsu, Japan) and the Image 1 STM marketed by the Karl Storz GmbH & Co (Tuttlingen, Germany) have been the most commonly employed devices in children [12].

To date, ICG-based FGS has been increasingly adopted in the field of pediatric urology with very promising results (Table 1).

Selective arterial mapping using ICG first proved to be valuable in six pediatric robot-assisted laparoscopic heminephrectomies for duplex kidneys in the series recorded by Herz et al. [17]. The real-time delineation of the vascular supply of both kidney moieties was performed in safety without bleeding or vascular complications. In this context, the ICG administration helped surgeons in identifying vascular anatomic variants and possible iatrogenic injuries to the remaining kidney pole, thus adding invaluable information during the minimally invasive procedure.

In 2019, Fernández-Bautista et al. [18] reported their first experience with ICG in three pediatric laparoscopic procedures involving the urogenital tract. In their series, an ICG-guided varicocelectomy and two ICG-guided nephrectomies were performed in children. During the laparoscopic Palomo varicocelectomy, the intravenous injection of ICG allowed for the observation of the spermatic artery first and then the venous vessels after. The ligation of the spermatic cord was therefore performed in safety while sparing the lymphatic vessels. During the two retroperitoneal laparoscopic nephrectomies, the intravenous injection of ICG provided a clear identification of the renal vascular anatomy and facilitated a safe dissection of the renal hilum.

The same year, Esposito et al. [19] reported an alternative use of ICG in 25 lymphatic-sparing laparoscopic Palomo varicocelectomies. Following the intratesticular injection of ICG, the lymphatic vessels were easily recognized and spared, and then the entire spermatic cord was clipped and divided.

In a following publication, the same authors [20] described their preliminary experience with FGS during 30 laparoscopic varicocele repairs, 3 laparoscopic nephrectomies, and 2 laparoscopic partial nephrectomies. The ICG solution was injected intravenously in all cases except for the varicocelectomies, in which case it was injected directly into the testis. The results of the study showed that ICG-FGS was a safe, useful, and versatile real-time imaging modality to provide a clear identification of the renal vascular perfusion and testicular lymphatic drainage in children.

Moreover, in a comparative analysis matching 57 ICG-guided procedures (41 varicocele repairs, 9 partial nephrectomies, 3 nephrectomies, and 4 renal cyst deroofings) with analogue standard surgeries, ICG proved to be highly beneficial in decreasing the operative time and the complication rate [21].

To go further, a more recent study tried to standardize ICG use in 22 laparoscopic partial nephrectomies [22]. While at the start of the operation ICG was injected into the ureter to identify the ureteral course (Figure 2), during surgery it was administered intravenously: a first time to identify the vasculature of the renal hilum and the non-functioning moiety and a second time, after the ligation of the vessel supplying the non-functioning kidney moiety, to identify the boundary plane between the avascular and the perfused pole.

Further uses of ICG technology in pediatric urology include the assessment of ureteral perfusion. In this regard, Carty et al. [23] described three robotic reconstructive cases (one congenital ureteral stricture, one mid-ureteral polyp disease, and one distal ureteral polyp disease) in which ICG was used to confirm a satisfactory blood supply after the excision of the affected ureteral segment at the end of the surgical anastomosis.

Moreover, the lack of bilitranslocase (a carrier protein of ICG present in normal proximal tubule cells) in renal tumors makes ICG a useful tool for tumor margin definition in pediatric surgical oncology. In this regard, Abdelhafeez et al. [24] reported that the intravenous injection of ICG the day before surgery optimized the tumor-to-background ratio and enhanced the tumor visualization in seven children with Wilms tumors and one child with an epithelioid angiomyolipoma. A prospective clinical trial enrolling 312 children and adolescents with various solid tumors is currently underway to establish the utility of this novel imaging tool in pediatric oncology [25].

Although the past few years have no doubt witnessed significant advances in FGS applications in pediatric urology, further preclinical developments and prospective clinical studies involving larger cohorts of patients are highly recommended to improve patient outcomes while minimizing perioperative complications.

Although only NIR-I dyes (excitation wavelength: 700–900 nm), such as ICG, have been approved for clinical use in biomedical fluorescence optical imaging, NIR-II fluorophores (excitation wavelength: 1000–2000 nm) have been shown to provide higher contrast, greater definition, and improved penetration depths, owing to less tissue autofluorescence and absorbance [12]. In this scenario, the discovery that some NIR-I dyes have displayed bright emission tails over 1000 nm offers appealing opportunities for FGS applications in humans.

Moreover, to maximize the signal from tumor cells and to minimize background noise, tumor-targeted fluorescent probes made of fluorescently labeled antibodies are currently under investigation in pediatric surgical oncology [10,12].

## 3. Three-Dimensional (3D) Reconstructions and Printing Technologies

Recent years have witnessed unpreceded research in the field of precision medicine in a move towards customized treatments and interventions based on each patient-specific anatomy and disease findings.

Novel technological advances and state-of-the-art biomedical research are turning this concept into reality for the new generation of surgeons.

Three-dimensional (3D) reconstructions and printing technologies are increasingly adopted in pediatric urology for surgical training, parent counseling, preoperative planning, and intraoperative navigation, thanks to their capability in translating radiological images into tangible replicas of the patient-specific anatomical details (Figure 1b).

In this regard, in 2014, Cheung et al. [26] developed a pediatric pyeloplasty simulator using 3D printing and silicone modeling for laparoscopic training and skills acquisition, while, in 2020, Ruiz et al. [27] used 3D-printed organs, inner mesh, and synthetic gel to train fellows for laparoscopic pyeloplasty, extravesical ureteral reimplantation, Mitrofanoff appendicovesicostomy, and uretero-ureteral anastomosis.

Amongst all the pediatric urological procedures, cloacal repair is one of the most demanding. Patients with cloacal malformations have a wide spectrum of anatomic variations that can affect the staged surgical reconstruction, eventually interfering with the long-term functional and cosmetical prognosis. Several imaging studies are therefore required preoperatively to properly plan the surgical procedure according to the patient-specific genitourinary anatomy. Over the past few decades, major strides have been taken forward in the reconstruction of this rare gastrointestinal and urological malformation to achieve optimal surgical outcomes.

In this regard, 3D reconstruction technologies and 3D printing techniques have been adopted to overcome some limitations owing to the traditional preoperative endoscopy, high-pressure distal colostogram, and non-contrast-enhanced MRI (Table 2).

In particular, in 2007, Baughman et al. [28] first introduced the use of 3D MRI technologies for the preoperative assessment of four females with cloacal malformations. In their study, the authors compared standard genitography, preoperative endoscopy, and 3D MRI genitography and concluded that the latter provided excellent anatomical details of the pelvic organs, levators, and lumbosacral spine without exposing the infants to ionizing radiation and added valuable complementary information to that of endoscopy.

Five years later, Patel et al. [29] described the use of 3D rotational fluoroscopy to obtain more precise anatomic details of the pelvic structures of infants with complex cloacal malformations, particularly regarding the length of the common channel and the appearance/location of the vagina and the bladder. The same year, Jarboe et al. [30] published the results of a case series involving two infants with complex genitourinary malformations in which 3D rotational fluoroscopy was combined with high-resolution 3D MRI to provide a very high definition of the lumen and precise measurements of channel/fistula lengths.

To go further, in 2017, Ahn et al. [31] compared 3D reconstruction cloacagrams with the endoscopic and intraoperative findings of four patients with cloacal malformations and used 3D printing technologies to create a 3D-printed model of the pelvic organs. The 3D reconstruction cloacagrams correlated well with the endoscopic and intraoperative findings, in particular with regards to the common channel length, the urethral length, and the level of the cloaca malformation, while the 3D-printed model showed potential applications for preoperative planning and the education of both families and trainees.

A year later, Gasior et al. [32] published a comparative analysis of four different cloaca imaging techniques aiming to advance the surgeon’s understanding of the complex pelvic anatomy in the case of cloaca malformations. According to the study design, a 2D contrast study cloacagram, a 3D model rotatable computer tomography (CT) scan reconstruction, a software-enhanced 3D video animation, and a printed physical 3D cloaca model of the same case of a cloaca patient were shown to 29 trainees in pediatric surgery and 30 consultants in pediatric surgery and urology. The results showed that the comprehension of the 2D imaging was the lowest and improved as the complexity of the imaging technology increased for both attendings and trainees. The authors concluded that the 3D reconstruction technologies and 3D-printed models enabled surgeons to make significant strides in the comprehension of complex cloacal malformations.

Finally, more recently, Krois et al. [33] tested the quality and feasibility of a real-size 3D-printed cloaca model for cysto-vaginoscopic evaluation for procedural training. Thirty-two pediatric surgeons and eight pediatric urologists were asked to perform a cysto-vaginoscopy on a real-size rubber-like 3D model of an infant pelvis with a cloacal malformation and to complete a brief questionnaire. Most participants rated the model as a valid training tool for real-life cases.

Besides cloacal malformations, the bladder–exstrophy–epispadias complex (BECC) represents the other greatest challenge for pediatric urologists today. It includes a broad spectrum of surgically correctable defects of the abdominal wall, bony pelvis, and urogenital tract, ranging from isolated epispadias to bladder and cloacal exstrophy [34]. In the latter, a correct understanding of the abnormal pelvic floor plays a decisive role in choosing the most appropriate surgical techniques to provide optimal long-term functional outcomes [34]. In this regard, the lack of ionizing radiation and the high soft-tissue and excellent spatial resolutions of preoperative 3D MRI have made it the technique of choice for the evaluation of the abnormal bony and musculoskeletal pelvic anatomy of these neonates [34]. Moreover, in women with BEEC, transperineal 3D ultrasonography has been used after surgical reconstruction for assessing biometric variables of the pelvic floor (i.e., levator hiatus diameter, hiatal area, levator angle, and pubovisceral muscle thickness) and predicting long-term functional outcomes [35,36].

Not only rare and complex urogenital malformations but also common congenital uropathies can benefit from the development of 3D mapping technologies. In this regard, a recent study published by Siapno et al. [37] demonstrated the excellent accuracy of structured light scanning, traditional photogrammetry, and photogrammetry with a 3D camera for angle assessments of 3D-printed inanimate blocks. The application of these innovative imaging systems in the surgical setting may, in the future, provide optimized parameters for chordee management and facilitate the objective analysis of cosmetic outcomes.

## 4. Augmented Reality (AR)

Augmented reality (AR) is the superimposition of digitally produced images, audio, and video over a user’s view to enhance his/her experience of the real world. In AR, the real-world environment is presented to the user but enriched and modified with computer-assisted applications, such as images, audio, and video (Figure 1c) [10,38].

AR has a broad area of applications in medicine, especially in surgery. With AR technologies, patient radiological images can be superimposed on the surgical field, allowing surgeons to better define the anatomical structures while remaining aseptic [10,38].

While these technologies have been becoming increasingly popular in adult urology, their adoption in pediatrics is still anecdotal [39].

In this regard, in 2013, Souzaki et al. [40] described an AR navigation system based on preoperative CT and MRI imaging for pediatric tumor resection. One Wilms tumor, one metastasis of rhabdomyosarcoma, one undifferentiated sarcoma, two bronchogenic cysts, and one hepatoblastoma were detected and successfully resected using an AR navigation system. In a more recent study, Wellens et al. [41] reported the use of AR and 3D printing technologies for optimizing the surgical planning of nephron-sparing surgery (NSS) in 10 children with Wilms tumors. The new preoperative 3D imaging strategy improved the surgeons’ understanding of the patient’s anatomy, enhanced preoperative surgical planning, and optimized the surgical procedure. Similar results were also reported by Chaussy et al. [42]. The authors investigated the role of 3D reconstructions of preoperative CT images in the surgical planning of 12 patients with Wilms tumors. They concluded that the 3D representation of the renal tumors and their vasculature provided valuable insight for choosing the most appropriate surgical technique by more precisely visualizing the tumor relationship with the intrarenal urinary cavities and the surrounding vessels. Moreover, the 3D reconstructions were particularly useful for the accurate measurement of the volume of the tumor and the surrounding healthy parenchyma, providing more precise selection criteria for NSS.

In the field of urolithiasis, despite the introduction of miniaturized instruments in percutaneous nephrolithotomy (PCNL) procedures, the percutaneous kidney puncture still represents the most challenging step [43,44]. This surgical procedure is characterized by the steepest learning curve due to the risk of damaging the surrounding blood vessels and organs, particularly in children.

In adult urology, Rassweiler et al. [45] described their clinical experience with an iPad^®^-assisted marker-based navigation for the percutaneous access to the kidney during PCNL. The superimposition of virtual markers onto the patient generated a computer-based algorithm able to define the correct access point and angle for PCNL. Subsequently, Müller et al. [46] used an iPad^®^ navigation system to puncture 53 kidneys, as performed by a urological trainee and two experts, and compared the puncturing time and the radiation exposure with ultrasound and fluoroscopy. With regards to the puncturing time, the trainee outperformed with the proposed AR system whereas the experts performed best with fluoroscopy. In terms of radiation exposure, the iPad^®^ assistance significantly lowered it for both the trainee and the experts.

Despite the AR devices and technologies being described for the surgical treatment of adult urolithiasis only, they also have the potential to revolutionize the surgical planning and treatment of urinary stones in children 40. Pediatric urologists need to keep up to date with these recent technological advances that will possibly reach the clinical practice soon.

## 5. Contrast-Enhanced Ultrasound (CEUS)

Contrast-enhanced ultrasound (CEUS) is a dynamic imaging technique that has been increasingly employed in pediatric urology. It provides morphological and functional data of the urinary tract by administering an ultrasonographic contrast agent, either intravesically or intravenously (Figure 1d) [47,48].

Compared to conventional radiological contrast studies, its main advantages are associated with the high quality of the images it provides without using ionizing radiation [41].

In pediatrics, the availability of ultrasound contrast agents made of stabilized microbubbles has favored the rapid development of CEUS for vesicoureteral reflux (VUR) diagnosis [49].

In this regard, a recent systematic review and meta-analysis published by Yousefifard et al. [48] confirmed the undeniable diagnostic role of CEUS for VUR detection in childhood. In particular, the AUC, sensitivity and specificity for VUR detection were 0.97 (95% CI: 0.95, 0.98), 0.92 (95% CI: 0.86, 0.96), and 0.94 (95% CI: 0.95, 0.98) for the 1st-generation contrast agents and 0.97 (95% CI: 0.95, 0.98), 0.93 (95% CI: 0.86, 0.97), and 0.91 (95% CI: 0.86, 0.95) for the 2nd-generation contrast agents, unequivocally proving the role of CEUS as a radiation-free alternative to voiding cystourethrogram (VCUG) for VUR diagnosis in children.

However, its applications are now evolving so that the entire urinary tract can be assessed using CEUS [47]. In this regard, a review article recently published by Duran et al. [47] highlighted the significant role CEUS achieved not only for the real-time assessment of the voiding function in children but also for the identification of obstructive and nonobstructive anomalies of the urethra.

Besides VUR detection, CEUS has been used for the diagnosis of several other congenital and acquired uropathies in childhood, including refluxing megaureters, ectopic ureters, ureteroceles, bladder diverticula, kidney trauma, and renal cortical cysts [47,50]. Moreover, it has shown excellent accuracy and reliability for the recognition of posterior and anterior urethral valves, anterior urethral diverticulum, prostatic utricle diverticulum, urethral strictures, and urethral ectasia in children [47,51].

Today, CEUS has also become part of the diagnostic armamentarium of pediatric urologists for the intraoperative assessment of the intra- and extraluminal characteristics of the male urethra. For example, it has been used intraoperatively to characterize a urethral stricture and a dense spongiofibrosis in a 17-year-old boy, significantly aiding in surgical decision making [52].

More recently, in adults, advanced CEUS images have been integrated and reconstructed with standard CT and MRI for the real-time evaluation of focal renal lesions, exploiting the advantages and reducing the limits of each individual imaging modality [53]. The good safety profile and cost-effectiveness and the high accessibility and repeatability of this fusion imaging modality make it a very interesting technique for pediatric oncology radiology.

## 6. Intraoperative Magnetic Resonance Imaging (iMRI)

Intraoperative magnetic resonance imaging (iMRI) has been seen as a very promising technique for assisting in several surgical procedures. The multiplanar imaging capabilities and the high spatial and contrast resolution of iMRI make it a particularly useful tool for differentiating normal and pathological anatomical structures during surgery (Figure 1e) [11].

iMRI has been particularly beneficial in pediatric neurosurgery, where it proved to be a safe and effective imaging modality for increasing the extent of tumor resection and delineating the relationships with the surrounding relevant structures [11].

In the field of pediatric urology, Di Carlo et al. [54] first described the feasibility of iMRI-guided surgical reconstruction during the closure of 43 classic bladder exstrophies and 4 cloacal exstrophies. The iMRI technology was used to guide the surgical procedures and identify the fibers of the urogenital diaphragm and the thickened muscle attachments between the posterior urethra, the bladder, and the bony pelvis. Immediately before surgery, the intraoperative registration was conducted after preoperative planning using five anatomical landmarks. This data were reconstructed into a 3D volumetric rendering, serving as an anatomical roadmap to guide the dissection of the pelvic floor. It is worth noting that 100.0% accuracy was proven in all patients in the correlation of gross anatomical landmarks with the 3D iMRI-identified landmarks, and all patients had a successful reconstruction with no major perioperative complications. Ongoing studies of imaging guidance in BEEC repair were warranted, as the surgical techniques are improving over the years [55].

## 7. Conclusions

FGS and the novel imaging devices described here represent an evolutionary process in the integration of advanced imaging with surgical intervention and inspire new perspectives for pediatric urologists. Highly sophisticated systems of image guidance will allow pediatric urologists to operate with an unprecedented level of accuracy and precision, bringing the simultaneous monitoring of anatomy and function into surgery.

Further testing of FGS, 3D imaging technologies, and the other novel intraoperative devices presented here, together with the increasing use of MIS instruments, will facilitate their adoption in the years to come for the benefit of children.

## Figures and Tables

**Figure 1 ijerph-19-11194-f001:**
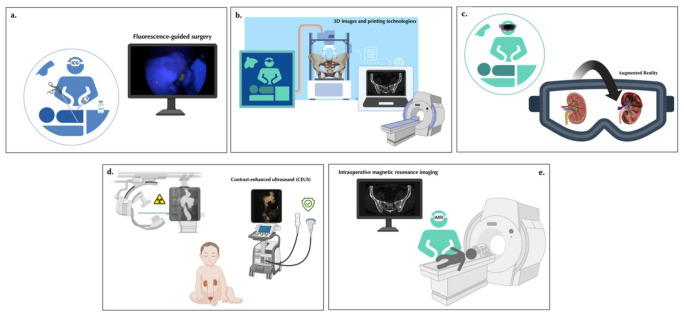
Schematic view of the novel innovative technologies for improved visualization in the field of pediatric urology described in this narrative review, including (**a**), indocyanine green (ICG)-based Fluorescence-guided surgery (FGS), (**b**) three-dimensional (3D) reconstructions and printing technologies, (**c**) Augmented reality, (**d**) Contrast-enhanced ultrasound (CEUS), and (**e**) intraoperative Intraoperative magnetic resonance imaging (iMRI).

**Figure 2 ijerph-19-11194-f002:**
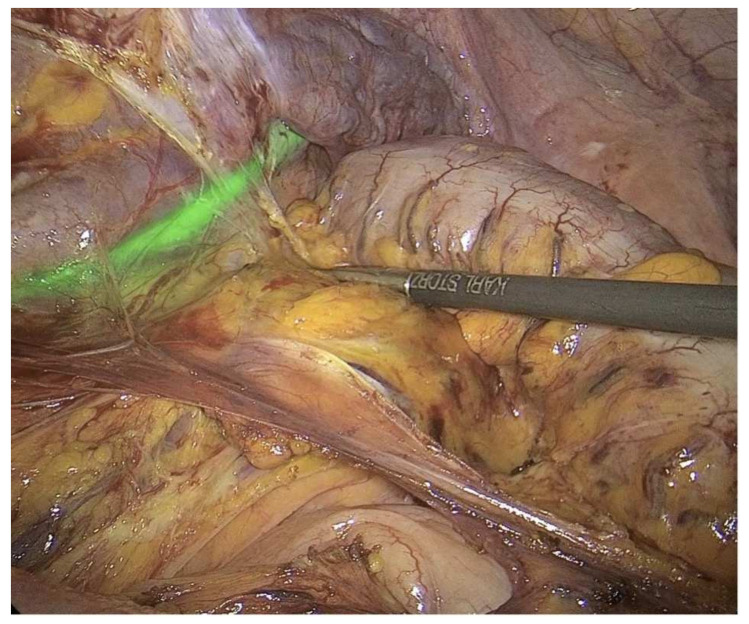
The intraoperative administration of indocyanine green (ICG) in the distal ureter can provide valuable insight during surgical dissection, preventing its unintentional damage (Courtesy of Professor Luigi Boni, Department of Surgery, Fondazione IRCCS Ca’ Granda Ospedale Maggiore Policlinico, University of Milan, Milan, Italy).

**Table 1 ijerph-19-11194-t001:** Selected scientific articles focusing on Fluorescence-guided surgery (FGS) in the field of pediatric urology.

Author, Year	Disease, Number of Patients	Type of Surgery	Dye (Administration Route and Dosage)	Imaging System	IDEAL Framework Stage
Herz et al. [17], 2016	Duplex kidney (*n* = 6)	Robot-assisted laparoscopic heminephrectomy	ICG (iv, 1.25–2.5 mg, 30–60 s prior to surgery)	Firefly^TM^ system, da Vinci (Intuitive Surgical, Inc., Sunnyvale, CA, USA)	2a
Fernàndez-Bautista et al. [18], 2019	Varicocele (*n* = 1); Non-functioning kidney (*n* = 2)	Laparoscopic Palomo varicocelectomy; Laparoscopic nephrectomy	ICG (iv, nd, at the time of the laparoscopic Palomo varicocelectomy; iv, 0.2 mg/kg, at time of thelaparoscopic nephrectomy)	1488 HD 3-Chip camera system, Stryker	1
Esposito et al. [19], 2019	Varicocele (*n* = 25)	Laparoscopic Palomo varicocelectomy	ICG (intratesticular, 10 mg, at time of surgery)	nd	2a
Esposito et al. [20], 2019	Varicocele (*n* = 30); Non-functioning kidney (*n* = 3); Duplex kidney (*n* = 2)	Laparoscopic Palomo varicocelectomy; Laparoscopic radical nephrectomy; Laparoscopic partial nephrectomy	ICG (intratesticular, 6.25 mg, at the time of thelaparoscopic Palomo varicocelectomy; iv, 0.5 mg/kg, at the time of the laparoscopic radical nephrectomy; iv, 0.3 mg/kg, at the time of the laparoscopic partial nephrectomy)	IMAGE1 S system (KARL STORZ SE & Co., KG, Tuttlingen, Germany)	2a
Esposito et al. [21], 2020	Varicocele (*n* = 41); Non-functioning kidney (*n* = 3); Duplex kidney (*n* = 9); Renal cysts (*n* = 4)	Laparoscopic and robotic Palomo varicocelectomy; Laparoscopic radical nephrectomy; Laparoscopic partial nephrectomy; Robotic renal cyst deroofing	ICG (intratesticular, 6.25 mg, at the time of thelaparoscopic Palomo varicocelectomy; iv, 0.3 mg/kg, at the time of the laparoscopic radical nephrectomy; iv, 0.3 mg/kg, at the time of the laparoscopic partial nephrectomy; iv, 0.3 mg/kg, at the time of the laparoscopic radical nephrectomy; iv, 0.3 mg/kg, at the time of the robotic deroofing of simple renal cysts)	Storz D-light (KARL STORZ SE & Co., KG, Tuttlingen, Germany) and Firefly^TM^ system, da Vinci Xi robotic platform (Intuitive Surgical Inc., Sunnyvale, CA, USA)	3
Esposito et al. [22], 2021	Duplex kidney (*n* = 12)	Laparoscopic partial nephrectomy	ICG, 1st step: 25 mg into the ureteral catheter just before surgery to identify the ureter; 2nd step: 0.3 mg/kg, iv, to identify the hilar vessel and the vasculature of the non-functioning pole during surgery; 3rd step: 0.3 mg/kg, iv, to identify the boundary plane between the avascular and the perfused pole after ligation of the vessel supplying the non- functioning moiety	nd (KARL STORZ SE & Co., KG, Tuttlingen, Germany) and ICG-NIRF RUBINA^TM^ system (KARL STORZ SE & Co., KG, Tuttlingen, Germany).	3
Carty et al. [23], 2021	Congenital ureteral stricture (*n* = 1); Mid-ureteral polyp disease (*n* = 1); Distal ureteral polyp disease (*n* = 1)	Robotic Heineke–Mikulicz strictureplasty (*n* = 1); Robotic ureteroureterostomy (*n* = 2); Robotic right lower pole ureterocalicostomy (*n* = 1)	ICG (iv, 0.086 mg/kg at the time of the robotic Heineke–Mikulicz strictureplasty; iv, 0.039 mg/kg at the time of the robotic ureteroureterostomy in the 2nd patient; iv, 0.067 mg/kg at the time of the robotic ureteroureterostomy in the 3rd patient; 0.046 mg/kg, at the time of the robotic right lower pole ureterocalicostomy in the 3rd patient)	Firefly^TM^ system, da Vinci Xi robotic platform (Intuitive Surgical Inc., Sunnyvale, CA, USA)	1
Abdelhafeez et al. [24], 2021	Wilms tumor (*n* = 7); Epithelioid angiomyolipoma (*n* = 1)	Bilateral nephron-sparing surgery (*n* = 3); Unilateral radical nephrectomy and concurrent contralateral nephron-sparing surgery (*n* = 1); Unilateral nephron-sparing surgery (*n* = 4)	ICG (iv, 1.5 mg/kg, 24 h before surgery)	Iridium system (Visionsense Corp, Philadelphia, PA)	2a

**Table 2 ijerph-19-11194-t002:** Selected scientific article focusing on 3D reconstruction technologies and printing techniques for cloaca repair.

Author, Year	Type of Study	3D Technology Developed	Aim of the 3D Technology Developed	Authors’ Conclusions
Baughman et al. [28], 2007	Prospective study comparing standard contrast genitography, endoscopy, and 3D MRI genitography for the preoperative surgical planning of four female infants with cloacal malformations	3D MRI genitography	Preoperative surgical planning	The 3D MRI genitography provided excellent anatomical details of the complex genitourinary anomalies, augmented the information obtained by standard MRI, and added complementary information to that of endoscopy.
Patel et al. [29], 2012	Review article describing the use of 3D rotational fluoroscopy for surgical planning and prognosis determination in children with complex cloacal malformations	3D rotational fluoroscopy	Preoperative surgical planning and prognosis determination	The 3D rotational fluoroscopy provided surgeons with detailed anatomical findings, enabling them to more accurately plan for surgical repair and predict functional prognosis.
Jarboe et al. [30], 2012	Case series on the use of 3D rotational fluoroscopy combined with high-resolution 3D pelvic MRI for the delineation of the pelvic anatomy of two female patients with complex genitourinary anomalies	3D rotational fluoroscopy combined with high-resolution 3D pelvic MRI	Preoperative surgical planning	The 3D rotational fluoroscopy combined with high-resolution 3D pelvic MRI provided excellent delineation of the pelvic anatomy to aid in operative planning.
Ahn et al. [31], 2017	Retrospective review comparing 3D reconstruction cloacagrams and endoscopic and intraoperative findings of four cloaca patients and reporting the use of 3D printing technology for preoperative planning and education	3D reconstruction cloacagrams and 3D-printed models of cloacal malformations	Parent counseling, surgical education for trainees, and preoperative surgical planning	The 3D reconstruction cloacagrams yielded accurate measurements of urethral length and level of cloaca common channel, consistent with the endoscopic findings. The 3D-printed models were useful for surgical planning and education.
Gasior et al. [32], 2019	Prospective study comparing a 2D contrast study cloacagram, a 3D model rotatable CT scan reconstruction, a software-enhanced 3D video animation, and a printed physical 3D model for preoperative planning of a cloaca malformation	3D model rotatable CT scan reconstruction; software-enhanced 3D video animation; 3D-printed physical cloaca model	Improve learning and understanding of cloaca malformations for preoperative surgical planning	The 3D reconstruction and printed models enabled surgeons to make significant strides in the comprehension of intricate cloacal anatomy and achieve a higher level of preparedness for surgery.
Krois et al. [33], 2021	Prospective study investigating the quality and the feasibility of a real-size rubber-like 3D model of an infant pelvis with a cloacal malformation for cysto-vaginoscopy	3D-printed cloaca model	Education, procedural simulation, and preoperative surgical planning	The 3D-printed cloaca model was useful for preoperative training to enhance the understating of the patient-specific pelvic anatomy

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
