# Peer review of "Fluorescence-Guided Surgery and Novel Innovative Technologies for Improved Visualization in Pediatric Urology"

_ijerph, 2022, doi:10.3390/ijerph191811194_

Round 1

Reviewer 1 Report (Previous Reviewer 2)

The review paper of Paraboschi Irene et al., Fluorescence-guided surgery and novel intraoperative optical imaging for enhanced visualization in pediatric urology, is well-written, comprehensive, and fluid. The number of references is adequate for a review paper that is not a systematic review of the literature. The tables are well-composed and clarify the examples from the literature. In general, English is well-written and the style is appropriate.

For a more clear understanding of the reader, the authors are advised to include more figures since this is a review paper. I suggest that each section from 2 to 6 should include one figure that best describes the section's topic. Authors can include a composed figure of examples from the literature or create their own images.

Author Response

Dear Reviewer,

many thanks for your kind comments. We deeply appreciate your feedback.

This article is a thorough and comprehensive narrative review of the use of fluorescence-guided surgery (FGS), 3-dimensional (3D) reconstructions and printing technologies, augmented reality (AR), contrast-enhanced ultrasound (CEUS), and intraoperative magnetic resonance imaging (iMRI) in the field of pediatric urology.

As you suggested, we have added a figure for each topic aiming to provide a clearer understanding for the readers. We have introduced a schematic overview of each technology described in this narrative review (Figure 1). Moreover, in Figure 2, we have attached an intraoperative picture kindly provided by a collaborator of ours. We decided not to include other published images to avoid copyright-related issues. 

We look forward to your comments and suggestions regarding our re-submission.

Yours sincerely,

Irene Paraboschi, MD

Department of Pediatric Urology

IRCCS Cà Granda Ospedale Maggiore Policlinico Milano, Italy

Reviewer 2 Report (Previous Reviewer 3)

The revisions and the added subsections improve the quality of manuscript and the tables add interests to the review content.

page 5:  correct "excitation wavelength" to replace by emission wavelength between 700-900nm and then between 1000-2000nm

minor: Add legend to the tables and in the corrected manuscript, the figure 1 appears deleted with no legend although they mentioned that the legend was rewritten in the reply to reviewer (?)

Author Response

Dear Reviewer,

Our group would like to thank you for the opportunity to resubmit a revised version of the manuscript entitled: ‘Fluorescence-guided surgery and novel intraoperative optical imaging for enhanced visualization in pediatric urology’.

A point-to-point reply to your comments is enclosed below. Attached you find a thoroughly revised version of the manuscript with the highlighted changes (in red) from the original version.

We thank you all very much for the useful comments, which have certainly improved the quality of our manuscript, and we hope it will now be acceptable for publication in this prestigious Journal.

We look forward to your comments and suggestions regarding our re-submission.

Yours sincerely, 

Irene Paraboschi, MD

Department of Pediatric Urology

IRCCS Cà Granda Ospedale Maggiore Policlinico Milano, Italy

Round 2

Reviewer 1 Report (Previous Reviewer 2)

The revised version of the manuscript can be suitable for publication in the IJERPH if the authors provide the figures in the manuscript and if I agree they are proper for publication.

The manuscript does not have the figures, so send a new version with the figures for appreciation.

This manuscript is a resubmission of an earlier submission. The following is a list of the peer review reports and author responses from that submission.

Round 1

Reviewer 1 Report

Summary 

The authors described several novel approaches in pediatric urology including the use of Fluorescence guided surgery (FGS) used to visualize normal and abnormal anatomical structures using indocyanine green (ICG) and tricarbocyanine. The authors also described the use of sophisticated cameras -The Photodynamic EyeTM marketed by Hamamatsu Photonics Co (Hamamatsu, Japan) and the Image 1 STM marketed by the Karl Storz GmbH & Co (Tuttlingen, Germany) to enhance the visualization of anatomical structures. 

A brief synopsis of the urological surgeries carried out was described by the authors, this includes varicocele repairs, partial nephrectomies, and renal cyst deroofing, etc. 

The authors also described other novel methods in pediatric urological surgery imaging techniques 

3. -dimensional (3D) reconstructions and printing technologies

Augmented reality (AR)

Contrast-enhanced ultrasound (CEUS)

Intraoperative magnetic resonance imaging (iMRI)

Strength 

A well-written description of novel and innovative techniques in pediatric urology 

Very descriptive and detailed for the most part

Weakness

The authors did not include images for

·        3. -dimensional (3D) reconstructions and printing technologies

·        Augmented reality (AR)

·        Contrast-enhanced ultrasound (CEUS)

·        Intraoperative magnetic resonance imaging (iMRI)

A review article at the most. Not an original research manuscript

Reviewer 2 Report

The article written by Irene Paraboschi et al. cannot be accepted for publication in the International Journal of Environmental Research and Public Health. Although the article is well-written, fluid, and organized, it lacks substantial literature in each section. More references could have been added to enrich the text because the number of references (only 44) is very limited for a review paper in the area of fluorescence-guided surgery for pediatric urology.

The abstract contains references to literature which is not acceptable. The authors should remove those references.

The Introduction is very incomplete and does not contain any references, which limits the range and overview of the article.

It would be recommended that the authors could have provided more figures (e.g. one per section), and a table summing up the articles in each section and highlighting the most important aspects of each one.

There are also some minor errors that I would like to mention:

- The references of the IJERPH are written in the form [Ref] and not only superscript as the authors did (Ref).

- The authors should write et al. instead of "et al".

Reviewer 3 Report

The title of the manuscript announces optical imaging techniques which means Imaging based on the interaction of light (photons) with the tissue. However, in the manuscript, the authors described printing technology, US imaging and MRI imaging which are not optical imaging technologies. The authors must change the title or remove all the paragraphs on non-optical imaging technologies. Only FGS part corresponds to the topic of the title.

The manuscript can’t be considered as a narrative review because the descriptions are presented in a somewhat simplistic manner. It’s more a short overview. We need information on dose for ICG injection? More data on specific dyes? Phase? Timing for imaging with each technology? Tolerance? Safety? Clinical studies’ results?.... The manuscript lacks tables with clinical data for each technology. Advantages are not all reported, and disadvantages of the technologies are not described.  Comparison between technologies? They list previous publications without description and discussion on their content.

The future adoption of the technologies for clinical work flow are not analyzed by the authors.

The authors have experience as shown by the image in Figure 1 but they don’t report their learning curve  and self-experience in the text.

Legend of the Figure 1 must be rewritten “Scheme 13. described 3 robotic ureteral reconstructive cases (1 congenital ureteral stricture, 1 mid-ureteral polyp disease, and 1 distal-ureteral polyp disease) in which ICG was employed to confirm a satisfactory ureteral blood supply after the surgical anastomosis”?.

As a review, the authors must be more exhaustive in the description of the basis of optical imaging. As example, for FGS, they must define the fluorescence term.

Other concern: add references in introduction.